# Zombie Firms during and after Crisis

**Ivana Blažková \***  **and Gabriela Chmelíková**

Faculty of Regional Development and International Studies, Mendel University in Brno, Zemědělská 1, 61300 Brno, Czech Republic; gabriela.chmelikova@mendelu.cz
* Correspondence: blazkova@mendelu.cz

**Abstract:** The phenomenon of zombie firms is gaining the attention of economists across different countries of the world; the increased interest is particularly evident after periods of economic crises. In our study, we focus on the development of zombie firms in the period before and after the 2008 crisis within two different economies, i.e., Germany and the Czech Republic, to provide insight into how different conditions and the overall economic context affect the fact that companies are more prone to becoming zombie firms. We implemented a difference-in-differences regression model to estimate the treatment effect by comparing the change (difference) in the differences in observed outcomes between these two countries. The data were obtained from two databases—the database Albertina by Bisnode a.s. providing financial statements of enterprises in the Czech Republic, and the database provided by Creditreform AG, which includes annual report data for a large sample of German companies. The dataset of German enterprises included 1,444,698 observations, i.e., 338,923 firms, and the dataset of Czech enterprises included 2,139,462 observations, i.e., 523,542 firms, both across the years 2000–2016, i.e., the data sample covered the period before and after the 2008 crisis. The different development of the share of zombie firms after the great financial crisis between Germany and the Czech Republic was proven as statistically significant. The findings confirm Germany is a country with a more stable economy and with a significantly lower risk of zombie firms' persistence, while the Czech Republic is at the level of the European average in terms of zombie share. The results also suggest an influence of post-crisis monetary policy on companies and the possible link between low interest rates and a growing share of zombies.

**Keywords:** zombie firms; crisis; Germany; Czech Republic; interest rate

## 1. Introduction

The term "zombie firm" is increasingly gaining the attention of economists across different countries of the world. Zombie companies are generally considered to be companies in financial distress—they usually report negative profits for a long time, they are heavily indebted and have problems meeting their interest obligations, yet they continue to operate, despite having lost all their equity (Mohrman and Stuerke 2014). Firms may become zombies through multiple channels—both external environment (i.e., macroeconomic effects such as business cycle fluctuations or changes in interest rates, and industry conditions, such as level of market concentration and competition) and internal factors (i.e., firms' behavior such as managerial decisions, debt policy, and other firm activities). Their presence in the economy appears to create certain negative externalities by crowding out the growth and expansion of healthy (non-zombie) companies (Banerjee and Hofmann 2018). The growth of healthy businesses is thus limited, together with the increased creation of barriers for start-ups (Caballero et al. 2008). Moreover, Rodano and Sette (2019) also point out that production factors are used in inefficient firms, which reduces the allocation efficiency of the economy.

Economists and scholars are increasingly addressing this issue (e.g., Kwon et al. 2015; Broz and Ridzak 2017; Banerjee and Hofmann 2018; Carreira et al. 2021; Copus et al. 2022) with the specific focuses of existing research on bank behavior known as "zombie lending"

or "evergreening" (see, e.g., Caballero et al. 2008; Shimizu 2012; Acharya et al. 2019), on the emergence and role of zombie firms in the context and countries (see, e.g., Urionabar-renetxea et al. 2016, 2018; De Martiis and Fidrmuc 2017; Blažková and Dvouletý 2022), and on the causes of zombies' survival (see, e.g., Iwaisako et al. 2013; Asanuma 2015); however, current developments show that interest in the existence and persistence of these companies in the market is renewed especially after periods of economic crises. It turns out that a certain development, usually a major economic event, can turn healthy firms into zombies. This was firstly evidenced by studies conducted by Japanese economists investigating problems related to the Japanese economic stagnation early after the year 2000 (e.g., Ahearne and Shinada 2005; Caballero et al. 2008; Kwon et al. 2015) and later by researchers from other countries dealing with the effects of the global financial crisis in 2008 on the phenomena of zombie firms (e.g., Broz and Ridzak 2017; Banerjee and Hofmann 2018; Carreira et al. 2021). At present, other publications highlight the risk of the proliferation of zombie firms in connection with the economic crisis caused by the COVID-19 pandemic (e.g., Zoller-Rydzek and Keller 2020; Dursun-de Neef and Schandlbauer 2021), which is an important impetus for realizing the need for this research. Unlike previous studies (e.g., Fischer 2021; Blažková and Dvouletý 2022), we are interested in the impact of the wider economic environment on the emergence and existence of zombie companies as a result of a crisis.

In every crisis so far, the economies of all countries, and in particular their financial systems, have faced various challenges; however, what turns out to be common is the lack of equity and excess debt in a growing number of firms (Ellul et al. 2020), which is becoming an important obstacle to the recovery of economic growth. As such, the issue of zombie companies is becoming very relevant. Given that this is not the first time we have encountered this situation, and as it is said, "lessons need to be learned from the past," it would be useful to be inspired in those countries where the recovery of companies to normal took place quickly after the crisis subsided. We should be interested in what has helped this recovery, whether there are any appropriate policies that should be put in place, and what environment is exacerbating the situation.

Therefore, in our study, we focus on the development of unproductive firms having problems with over-indebtedness, i.e., zombie firms, in the period before and after the 2008 crisis, which may provide interesting insights into possible developments after the current crisis caused by the COVID-19 pandemic. Using the example of two different economies, the results attempt to provide a benchmark of the zombie share of the Czech and German economies. While Germany is believed to be among the most stable economies in the European Union and is considered the leading country in the EU in monetary and financial problem solving (Kickert 2012), the Czech Republic, as a Central European country, underwent a process of transformation in the 1990s and the functioning of market mechanisms had to be implemented subsequently. The comparison of zombie firm distribution between these two economies should provide insight into how different conditions and the overall economic context affect the fact that companies are more prone to becoming zombie firms.

The paper proceeds as follows. The second part reviews theoretical and empirical studies that focus on the factors influencing the zombie firms' population before, during, and after a crisis, and develops our research question and hypothesis. The third part describes our data and methods. The empirical study presented in the fourth part focuses on the relationship between the share of zombie firms in the Czech Republic and Germany before and after the financial crisis in 2008. The concluding part discusses our main results and gives recommendations for policymakers regarding the fact that different conditions and environments require different approaches to solving this problem.

## 2. Effects of Monetary Policies on Zombie Lending—Literature Background and Hypothesis Formulation

The population of zombie firms differs from country to country and is driven by several factors that include both institutional and country-specific drivers related to economic

policy. In particular, the mechanism of monetary policy that central banks use to mitigate the effects of crises on zombies born is not systematically explored in the literature. This paper aims to fill this research gap by exploring the influence of two different monetary policies within the European area.

The behavior of the zombies' population during crises became an object of several empirical studies. Peek and Rosengren (2005) examined the allocation of credit in Japan and found out that firms during and after crises, when they face poor financial conditions, are more likely to receive additional bank capital. This behavior is also associated with banks' decreased capital ratios. Giannetti and Simonov (2013) explored the effects of credit size on the success of bank bailouts during the crisis. They came to the finding that a sufficiently large capital injection can increase the supply of credit and secure further investments for the firms. Ferrando et al. (2015) focused their research interest on the development of the European financial market after the great financial crisis in 2007–2009. They investigated the effect of stress on public finance and subsequent unconventional monetary policy launched by the European Central Bank. They documented that firms in the stressed countries were more likely to decrease their indebtedness, including debt securities, trade credit, as well as government-subsidized loans. On the other hand, firms with a healthy financial condition and credit history took advantage of the easier credit access. This situation widened the gap between prosperous and non-prosperous companies in Europe after the crisis. These findings were later confirmed by Acharya et al. (2019), who explored the effects of the European Central Bank's unconventional monetary policy after the great financial distress and confirmed that the action was not fully translated into economic growth. The firms receiving supported loans did not use them to invest but made reserves and banks remained undercapitalized. Storz et al. (2017) explored the relationship between firms' indebtedness and the financial health of both corporate and financial sectors. Their findings show that firms from stressed countries in the periphery of the Eurozone tend to increase their leverage compared to firms tiding to a strong bank sector.

Adalet McGowan et al. (2018) used the data covering the crisis years, i.e., 2003–2013, to investigate whether the rise of zombie firms could be partly a cyclical story. They argue that shocks that raise the prevalence of zombie firms can also adversely affect firm performance; however, these cyclical effects have proven not to be the cause, given the continued growth of zombies after the crisis (Adalet McGowan et al. 2018). The effect of the economic crisis was studied by Caballero et al. (2008), whose findings showed that the proportion of weakly profitable firms increases during periods of low interest rates and that the proportion of zombies increased, especially after the financial crisis.

The great financial crisis in 2007–2009 was accompanied by several negative manifestations in the form of a deterioration in the financial health of banks and a consequent deterioration in companies' access to capital. Countries, or monetary unions, have approached these problems differently (Goodhart 2008). In general, central banks have sought to keep reference interest rates low to encourage corporate investment and support economic recovery. Governments have focused their fiscal policies on increased government spending, which has led many countries to face excessive public debt (Spilimbergo et al. 2009).

As we aim to the comparison of zombie firms' distribution in the Czech Republic and Germany, two different monetary units are involved in our analysis. Germany represents a member of the Eurozone, with the European Central Bank in the lead. The Czech Republic, with its own currency, stands outside the Eurozone and has its own system of banking institutions headed by the central bank. Even though the Czech National Bank is a member of the European System of Central Banks, its decision-making powers are sovereign and monetary policy is thus different from that adopted by the European Central Bank. In the Eurozone, the great financial crisis was followed by a sovereign debt crisis, especially in periphery countries, which led the European Central Bank to take several measures to improve the situation in the banking sector and hence make the credits more accessible. The announcement of the Outright Monetary Transactions Program (OMT) in 2012 has belonged to the most important monetary policy employed in the euro area so far and

brought recapitalization of poorly capitalized banks, which passed these capital sources on firms, which might lead to a rise in the share of low-productivity, possibly non-viable firms with high levels of financial debt. The development of interest rates after the crisis in both monetary areas was also different. Compared to the European Central Bank, the Czech National Bank tried to keep interest rates higher, which made money more expensive, worsened access to credit, and hampered economic growth, but, on the other hand, prevented the nascency of zombie firms.

The starting point of our analysis is the comparison of the development of one of the basic monetary policy tools—the interest rate. We take the interest rate on the main refinancing operations, which is set by the Governing Council of the European Central Bank for the Euro area and by the Bank Board of the Czech National Bank for the Czech Republic. Figure 1 presents the development of this key instrument in both countries during the observing period—before, during, and after the great financial crisis.

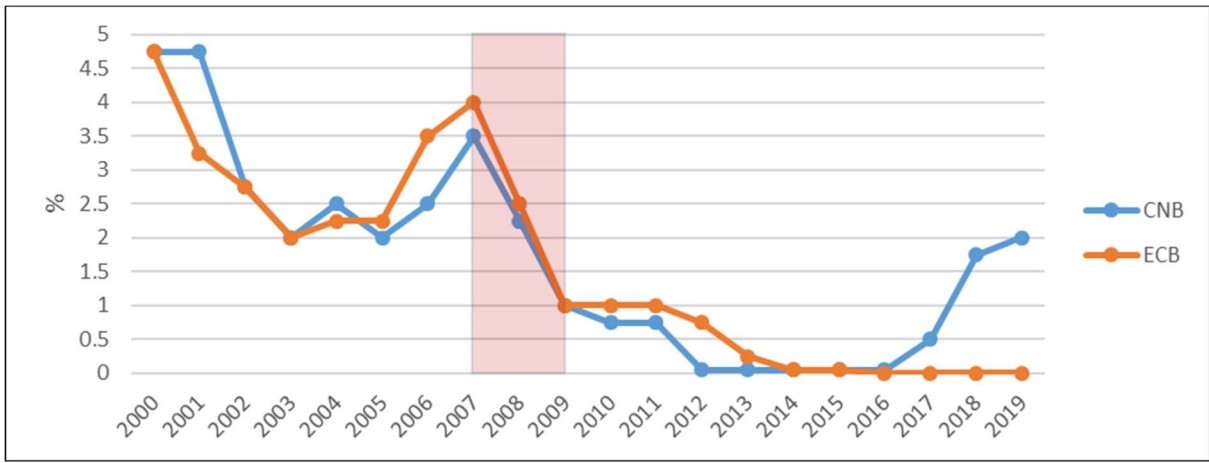

**Figure 1.** The interest rate (31 December) on the main refinancing operations in the Eurozone (ECB) and in the Czech Republic (CNB) (Source: The European Central Bank and the Czech National Bank).

The different approach of both monetary areas is clearly visible. Although before, during, and shortly after the great financial crisis, the European Central Bank kept interest rates above the level of the Czech Republic, the development later after the crisis was the opposite. Higher interest rates in the Eurozone made money more expensive and might prevent the high indebtedness of German firms, which could subsequently enter the financial crisis in better condition. Furthermore, lower interest rates after the crisis and the OMT program have offered a temporary lifeline for many overstretched companies in Eurozone. Our interest, therefore, is to answer the research question to what extent the different approach of both countries influences the scale of negative equity companies after the crisis; we accordingly propose our hypotheses as follows:

**H0.** *There is not a difference in the development of the share of zombie firms after the great financial crisis between Germany and the Czech Republic.*

**H1.** *There is a difference in the development of the share of zombie firms after the great financial crisis between Germany and the Czech Republic.*

### 3. Materials and Methods

*3.1. Research Design and Variables*

Before testing our hypothesis, we focused on how significant the existence of zombies within observed economies is, on differences between the countries, and the relationship of the development of zombie companies due to the crisis. To test our hypothesis, we implemented a difference-in-differences regression model used to estimate treatment effect by comparing the change (difference) in the differences in observed outcomes between

the treatment and control group (Angrist and Pischke 2008). The appropriateness of this method is evidenced by the fact that it is commonly used to assess the causal effect in the observed data where the experimental design is out of the researcher's control and usually subjected to unobserved confounders and some form of bias.

We estimate an econometric model, evaluating whether the different development of the share of zombie companies in Germany and the Czech Republic as a result of the crisis is statistically significant. Our focus on these countries is appropriate due to their mutual economic differences and conditions, which can provide significant insights into the occurrence and development of zombie companies depending on different economic contexts. Large differences in the post-crisis development of German and Czech companies lead to the hypothesis of different treatment of these two groups of companies at that time. If one group of companies undergo treatment, this should be shown in a difference-in-differences regression model. Considering how the Czech firms were treated, the dummy variable CZECH takes the value 1 for all Czech firms and the value 0 for all German firms (as commonly used, see, e.g., Secinaro et al. 2021). We use a dummy variable indicating the time when the treatment started. The treatment is expected to begin in 2009; therefore, the dummy variable TIME takes the value 0 in pre-2009 years and the value 1 in 2009 and post-2009 years. Additionally, an interaction term DID is created between CZECH and TIME. We estimate the simple difference-in-differences regression model for pairs $(x_j; Y_j)$, $j = 1, \ldots, n$, with intercept $\beta_0$ and slope $\beta_1$:

$$ZOMBIE_{SHARE} = \beta_0 + \beta_1 CZECH + \beta_2 TIME + \beta_3 DID + \varepsilon_j, \tag{1}$$

where $\varepsilon_j$ is a normally distributed random deviation with mean 0 and variance $\sigma^2$; that is, $\varepsilon_j \sim \mathcal{N}(0, \sigma^2)$ for all $j$. This approach makes it possible to determine whether the trends observed in the data can be confirmed by statistically significant treatment of Czech companies in comparison with German companies since 2009.

*3.2. Data Collection*

To collect the data for investigation, we employed two databases—the database Albertina by Bisnode a.s. providing financial statements of enterprises in the Czech Republic, and the database provided by Creditreform AG, which includes annual report data for a large sample of German companies. The analysis was implemented considering the years 2000–2016 to cover the period before and after the 2008 crisis. Thus, our study uses historical secondary data, which are readily available and are free from biases (Secinaro et al. 2020). The dataset of German enterprises included 1,444,698 observations, i.e., 338,923 firms across the years 2000–2016. The dataset of Czech enterprises included 2,139,462 observations, i.e., 523,542 firms across the years 2000–2016. The sample is not limited to a specific sector. Within both datasets, zombies were identified as firms reporting negative equity (i.e., the debts of that firms exceed their assets); therefore, we focus on the most extreme type of zombie firms as previously identified by Urionabarrenetxea et al. (2016, 2018) or Blažková and Dvouletý (2022). The number of firms in total and the number of zombies in individual years in both countries are presented in Table 1.

**Table 1.** Dataset structure concerning zombie/non-zombie classification (Source: Bisnode a.s., Creditreform AG; authors' own elaboration).

| | Germany | | | Czech Republic | | |
|---|---|---|---|---|---|---|
| | Observations in Total | Zombies | Share of Zombies in % | Observations in Total | Zombies | Share of Zombies in % |
| 2000 | 6334 | 230 | 3.63 | 44 | 6 | 13.64 |
| 2001 | 14,999 | 407 | 2.71 | 387 | 40 | 10.34 |
| 2002 | 29,302 | 891 | 3.04 | 5866 | 1014 | 17.29 |
| 2003 | 47,985 | 1610 | 3.36 | 59,929 | 13,633 | 22.75 |
| 2004 | 64,162 | 2290 | 3.57 | 77,610 | 17,305 | 22.30 |
| 2005 | 88,045 | 2899 | 3.29 | 92,110 | 19,441 | 21.11 |
| 2006 | 99,338 | 3336 | 3.36 | 108,314 | 21,729 | 20.06 |
| 2007 | 90,691 | 3012 | 3.32 | 128,890 | 24,849 | 19.28 |
| 2008 | 99,306 | 4040 | 4.07 | 148,686 | 28,965 | 19.48 |
| 2009 | 106,327 | 5278 | 4.96 | 165,188 | 33,933 | 20.54 |
| 2010 | 111,564 | 5334 | 4.78 | 172,815 | 36,367 | 21.04 |
| 2011 | 116,682 | 5167 | 4.43 | 180,221 | 38,718 | 21.48 |
| 2012 | 119,347 | 4952 | 4.15 | 185,781 | 40,929 | 22.03 |
| 2013 | 118,870 | 5009 | 4.21 | 197,684 | 44,144 | 22.33 |
| 2014 | 121,013 | 5014 | 4.14 | 203,764 | 46,283 | 22.71 |
| 2015 | 114,069 | 4744 | 4.16 | 204,177 | 45,650 | 22.36 |
| 2016 | 96,664 | 3587 | 3.71 | 207,996 | 46,566 | 22.39 |

## 4. Results

In order to find out the extent of the problem and the differences in the monitored countries, we examined several indicators, both in absolute and relative terms. The analysis of the data for the period 2000–2016 shows that about 20% of the Czech companies have negative equity and, therefore, can be considered zombies. In contrast, the situation in Germany is significantly better; zombies make up about 3% of companies in the period 2000–2016. If we compare the values with the average in the European Union, i.e., 20% as reported by Urionabarrenetxea et al. (2016), it is possible to confirm Germany as a country with a more stable economy with significantly lower risk, while the Czech Republic is at the level of the European average.

As can be observed in Figure 2, the share of zombie companies in the German economy experienced a distinct development after the crisis compared to the Czech economy.

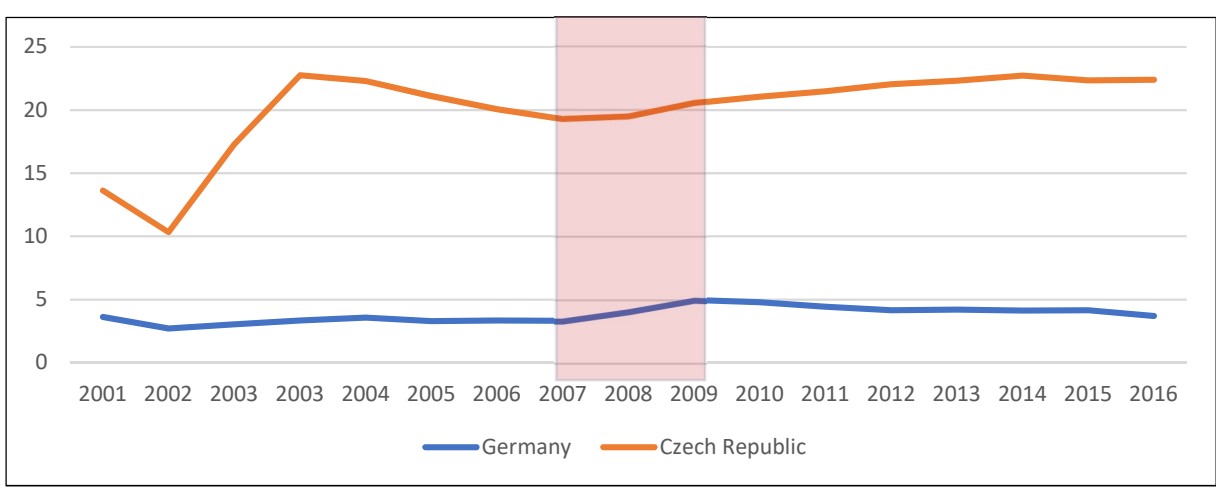

**Figure 2.** Share of zombie firms on total companies in %.

Should this different development be induced by a drop in interest rates in 2009, a connection between monetary policy and an increase in zombie firms at that time could

be established; therefore, a difference-in-differences analysis should provide statistical evidence and significance using Czech firms as the treatment group (*CZECH*) and German firms as the control group. The treatment supposedly began in 2009, as visible in Figures 1 and 2. From this time, the share of zombie companies in the Czech Republic shows a steady increase while the share of zombie companies in Germany experiences a steady decrease. The variable *TIME* indicates the years 2009 or later as the time of the treatment. The interaction term *DID* accounts for an interaction between the time and the group exposed to the treatment as *TIME* times *CZECH*. Table 2 displays the results of the difference-in-differences analysis.

**Table 2.** Model table.

|  | Dependent Variable Zombie |
|---|---|
| *CZECH* | 15.098 *** (1.380) |
| *TIME* | 0.946 *** (0.189) |
| *DID* | 2.445 * (1.413) |
| Constant | 3.372 *** (0.126) |
| Observations | 34 |
| $R^2$ | 0.942 |
| Adjusted $R^2$ | 0.936 |
| F Statistic (df = 3; 30) | 1386.942 *** |

Note: * $p < 0.1$; ** $p < 0.05$; *** $p < 0.01$.

All independent variables are statistically significant, at least at the 10% level. Most importantly, the coefficient for *DID* is significant at the 10% level, with the treatment having a positive effect. This leads to the conclusion that the treatment increases the zombie share within the Czech companies compared to German companies after the crisis in 2008. High R squared signals a good fit of the model; therefore, the explanatory value of the difference-in-differences analysis results is sufficient in this case. Based on the given statistical significance and based on the achieved results, the null hypothesis of the indifference assumption is rejected. We can hence support the hypothesis on different development of the share of zombie firms after the great financial crisis between Germany and the Czech Republic, which was proven as statistically significant.

## 5. Discussion

As follows from the estimated model, the findings suggest an influence of post-crisis monetary policy on companies and the link between low interest rates and a growing share of zombies. As demonstrated by previous research (e.g., Maddaloni and Peydró 2011; Delis and Kouretas 2011), banks are willing to bear a higher risk when interest rates are low, which leads to the softening of lending standards. Previous studies found a clear link between low interest rates and an increase in zombie share (Banerjee and Hofmann 2018), which is in coherence with the findings in Table 2. On the other hand, low interest rates may not be detrimental in resolving the crisis, as they can boost the economy. The result may be a slowdown in demand decline, which is important during a recession (Mian et al. 2015). So, there is no generally clear answer about low interest rates in the context of the crisis—lower rates will trigger aggregate demand as a positive effect on the economy, but an increase in the number of zombie companies will mean a misallocation of resources (Reid 2018).

Every crisis, whether we mean the 2008 crisis or the COVID-19 crisis, is a shock to the economy and means a decline in economic activity. Demand is falling, many companies are closing down, either due to bankruptcy or lockdown), and as a result, people are losing their jobs (Altavilla et al. 2020). To help businesses and also to protect jobs, governments are putting in place measures to support lending to businesses to cope with the crisis; however, is there not also support for zombie companies?

Banks should still keep sound lending standards, even during the crisis; however, if they, based on the public guarantees of government, start to lend more and even to companies that are in trouble due to the crisis, it may be more difficult to verify a client's creditworthiness, such as a financial situation or a risk of insolvency. There is therefore a risk that a loan granted to a non-viable company will be overlooked.

Ultimately, the findings of our study serve as an indication of the link between low interest rates and the increase of zombie firms in the economy. Although the policy of monetary loosening is in some cases justifiable and commonly used (Giles 2013), it can also help keep zombie firms alive.

## 6. Conclusions

The study aimed to provide a comparison of the zombie share of the Czech and German economies in the light of developments before and after the crisis in 2008. Based on the data from these different economies in 2000–2016, it can be said that there are statistically significant differences in the development of the zombie share after the crisis between these economies. While in the Czech economy, the share of zombies has been growing steadily since the beginning of the crisis, in Germany, the share of zombies has increased during the crisis, but after 2009 it has been gradually declining.

It should be borne in mind that any financial crisis will generate new zombie firms. This is not only because the economy is stagnating but also because governments will try to recover the economy from this situation and implement support not only for the banking sector but for the entire economic system, i.e., not only for healthy companies but also for non-viable ones (Stiglitz and Rashid 2020) since the widespread state aid cannot adequately distinguish between healthy companies and companies in poor conditions before the crisis; therefore, one of the consequent serious problems in the post-crisis period becomes the expansion of zombie firms in the economy, which should not be neglected in studies investigating the effects of a crisis and the potential for economic recovery. Given that zombie companies are a relatively new phenomenon, we would like to encourage researchers to take the above facts into account in their research. In addition to this note addressed to other researchers and scholars in order to broaden theoretical knowledge, we also bring the following practical implications regarding central bank policy and institutional arrangements.

After the crisis in 2008, central banks cut their interest rates to zero and, in some countries, even to negative levels to prevent the deepening crisis. It can be understood that if firms obtain a loan with a zero interest rate, they "cannot" go bankrupt; however, the side effect is the survival of firms that would not survive in a normal competitive market environment. As economists point out (e.g., Pikora 2019), lower interest rates by central banks support the economy, but at the same time, if interest rates are low for a long time, it creates new problems—the economy ceases to be efficient, preserving inefficient and non-viable activities; therefore, our research implies that central banks should have low rates only briefly so as not to create zombie companies, as it turned out to be effective even after the crisis in 2008 in Germany.

However, the problem is not only that the zombie share is growing but also the question of how to decrease the growth of these companies; therefore, our recommendation is the enactment of bankruptcy legislation for insolvent economic entities, as already pointed out by Papava (2020), since the use of bankruptcy laws seems to be an effective mechanism for eliminating zombie firms, i.e., as stated by Blažková and Dvouletý (2022), firms would be encouraged to generate positive cash flows allowing them to reduce debt and to restore the

"healthy" capital structure. Particularly in post-communist economies, such as the Czech Republic, the bankruptcy laws are not considered effective (Blažková and Dvouletý 2022). Bankruptcy should not always be perceived negatively because, in reality, it is an economically viable process in which unprofitable companies make way for the stronger ones.

Finally, we would like to draw attention to the fact that the current crisis due to the COVID-19 pandemic will exacerbate the adverse phenomenon of zombie firms. This is due to the environment of low or zero interest rates, which distorts the process of optimal capital allocation due to the huge inflow of newly created liquidity into financial markets and government rescue packages supporting companies across sectors, regardless of their productivity. Although these stimulus measures may prevent a significant economic downturn in the short term, the economic consequences can be expected to be much more serious in the long run.

Finally, several research limitations need to be mentioned. Since our study is based on corporate financial data, we were limited by the availability of data. Although we have used the largest and most comprehensive databases with the financial statements of companies in the analyzed countries, there are still companies, especially small ones, not required to disclose their financial results, which can skew the findings. It is also worth noting that there is a lack of prior research studies that would look at the impact of the crisis on the rise of zombie companies, whether it is a past crisis or the current crisis related to the COVID-19 pandemic; therefore, the conclusions drawn are not easy to compare with other countries and contexts, and conclusions may therefore be somewhat subjective. We would like to encourage researchers to further explore this growing and noticeable issue of zombie firms across various economies and contexts, which we see as an interesting and challenging research topic. Regarding further research, the importance of the social and institutional environment in the genesis and persistence of zombie firms should be examined, particularly their mutual relationship. We believe that more evidence is needed to clarify this relationship, especially from a wider range of countries and sectors of the economy.

**Author Contributions:** I.B. and G.C. worked together on the research—both authors conceived the overall baseline study and reviewed the existing studies, managed the data collection, performed the analysis, wrote the original draft preparation, and critically reviewed the manuscript. All authors have read and agreed to the published version of the manuscript.

**Funding:** This research was funded by AKTION ČR—Rakousko, grant number 90p2 (2021) and 93p5 (2022).

**Institutional Review Board Statement:** Not applicable.

**Informed Consent Statement:** Not applicable.

**Data Availability Statement:** Data used in this article were obtained from Creditreform AG (https://www.creditreform.de/; accessed on 16 September 2021) and Bisnode, a.s. (www.bisnode.cz; accessed on 30 September 2021).

**Acknowledgments:** The authors thank the anonymous referees for their contributions to the development of this article. The authors also thank the Czech National Agency for International Education and Research that supported our research under the AKTION ČR—Rakousko program (grant number 90p2 in 2021 and grant number 93p5 in 2022).

**Conflicts of Interest:** The authors declare no conflict of interest.

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
