# Peer review of "Zombie Firms during and after Crisis"

_jrfm, doi:10.3390/jrfm15070301_

Round 1

Reviewer 1 Report

The abstract is too short and not very informative. It does not reflect the body of the article well.

All figures and tables are clear enough to summarize the results for presentation to the readers. Figures and tables must be referred to in the text before appearing in the text. However, Figure 2 is referred to after and not before its appearance in the text.

The reference section is informative. However, not all references are accurate. The formatting of the references must be revised and corrected to make it more homogeneous and in accordance with the journal's rules.

The formatting of the entire text of the article, the references section, figures and tables should be improved.

Theoretical analysis is not very strong, it could be reinforced.

The approaches and methodologies used in the study are not new. The novelty lies in its application to a specific situation.

Author Response

Dear Reviewer,

Thank you very much for the time and efforts you invested in our manuscript, and we hope that we have been able to address all your comments. We have revised our manuscript according to your comments, and we explain the changes in light of the comments below. All changes in the manuscript are tracked.

Reviewer´s Comment:

The abstract is too short and not very informative. It does not reflect the body of the article well.

Response:

We agree with the reviewer that the abstract is short and some important information is missing, so we expanded it to better reflect the content of the paper.

Reviewer´s Comment:

Figures and tables must be referred to in the text before appearing in the text. However, Figure 2 is referred to after and not before its appearance in the text.

Response:

We thank the reviewer for pointing out this formal shortcomming of out manuscript. This incorrect reference is now fixed.

Reviewer´s Comment:

The reference section is informative. However, not all references are accurate. The formatting of the references must be revised and corrected to make it more homogeneous and in accordance with the journal's rules.

Response:

Thank you for pointing out the shortcomings in this section. We have corrected the references so that they are arranged alphabetically and in the correct format.

Reviewer´s Comment:

The formatting of the entire text of the article, the references section, figures and tables should be improved.

Response:

The manuscript was submitted in the journal template, therefore, it is not clear to us what text formatting the reviewer has in mind.

Reviewer´s Comment:

Theoretical analysis is not very strong, it could be reinforced.

Response:

We would like to thank the reviewer for commenting on this issue that motivated us to improve the quality of our manuscript. We have modified the introduction and theoretical background, which are now based on a broader discussion of current knowledge and publications.

Reviewer 2 Report

Dear authors, it is a pleasure for me to read in advance your contribution on the emerging topic of the so-called zombie firms. 

- Introductions are notoriously challenging to write. However, the first lines of your paper make very interesting reading. I will dwell a few lines on originality. After researching the primary databases for business and management studies (Scopus, WoS, Google Scholars), I consider the work original. However, I would urge you to distinguish it from the following work already at the introduction stage:
-Fischer, F. B. (2021). Zombie firms, corporate restructuring and relationship banking: Credit guidance as a key to tackle zombie lending? Accounting, Economics and Law: A Convivium, doi:10.1515/ael-2020-0065
- Blažková, Ivana, and OndÅ™ej Dvouletý. 2020. Zombies: Who are they and how do firms become zombies?. Journal of Small Business
Management: 1-27.
- Hallak, I., Harasztosi, P., & Schich, S. T. (2018). Fear the Walking Dead?: Incidence and Effects of Zombie Firms in Europe. Publications Office of the European Union.
 The differentiation from these would allow you to stand out for your originality.

- I appreciate the boldness of the authors in proposing solutions to the crisis caused by covid-19. However, as of today, it is uncertain when the pandemic situation will be overcome. It could be in 1 year, 5 years or 10 years. I consider this implication subjective. Therefore, I suggest you better emphasise the commonalities between the present economic crisis and the past one you referred to.

I suggest that the authors stress your study's theoretical and managerial implications as much as possible to make the introduction appealing to the reader.

- Although your methodology appears clear, I suggest you devote a few lines to research design. I recommend some contributions in the literature to strengthen this section:
- Secinaro, S., Brescia, V., Calandra, D. and Saiti, B. (2020), "Impact of climate change mitigation policies on corporate financial performance: Evidence-based on European publicly listed firms", Corporate Social Responsibility and Environmental Management, Wiley Online Library, Vol. 27 No. 6, pp. 2491-2501.
- Secinaro, S., Radwan, M., Calandra, D., & Biancone, P. (2021). Halal certification impact on firms' corporate social responsibility disclosure: Evidence from the food & beverage sector in Italy. Corporate Social Responsibility and Environmental Management, 28( 4), 1376- 1385. https://doi.org/10.1002/csr.2161

- I suggest providing the discussion section separately from the results. Thus, you can better link it with the satisfactory literature review you have provided. In fact, at this stage, the discussion appears very weak and does not give enough emphasis to your interesting results.

- I suggest you provide theoretical and practical implications in the concluding section. Although they emerge, it does not seem explicitly clear what they are.

- The conclusion lacks the research limitations. I would suggest you integrate them.

- I recommend expanding the section on future research fields, which is essential to continue the ongoing debate on the proposed topic.

Author Response

Dear Reviewer,

Thank you very much for the time and efforts you invested in our manuscript, and we hope that we have been able to address all your comments. We have revised our manuscript according to your comments, and we explain the changes in light of comments below. All changes in the manuscript are tracked.

Reviewer´s Comment:

Introductions are notoriously challenging to write. However, the first lines of your paper make very interesting reading. I will dwell a few lines on originality. After researching the primary databases for business and management studies (Scopus, WoS, Google Scholars), I consider the work original. However, I would urge you to distinguish it from the following work already at the introduction stage:

- Fischer, F. B. (2021). Zombie firms, corporate restructuring and relationship banking: Credit guidance as a key to tackle zombie lending? Accounting, Economics and Law: A Convivium, doi:10.1515/ael-2020-0065

- Blažková, Ivana, and OndÅ™ej Dvouletý. 2020. Zombies: Who are they and how do firms become zombies?. Journal of Small Business Management: 1-27.

- Hallak, I., Harasztosi, P., & Schich, S. T. (2018). Fear the Walking Dead?: Incidence and Effects of Zombie Firms in Europe. Publications Office of the European Union.

 The differentiation from these would allow you to stand out for your originality.

Response:

We would like to thank the reviewer for this constructive comment. Thanks to this comment, we have modified the introductory part to better reflect the difference of our study from previous publications, namely that we focus on the influence of the wider economic environment on the emergence and existence of zombie companies in the economy, particularly as a result of a crisis.

Reviewer´s Comment:

I appreciate the boldness of the authors in proposing solutions to the crisis caused by covid-19. However, as of today, it is uncertain when the pandemic situation will be overcome. It could be in 1 year, 5 years or 10 years. I consider this implication subjective. Therefore, I suggest you better emphasise the commonalities between the present economic crisis and the past one you referred to.

I suggest that the authors stress your study's theoretical and managerial implications as much as possible to make the introduction appealing to the reader.

Response:

We agree with the reviewer that the uncertainty about the development of the COVID-19 crisis is still high, making it impossible for us to draw any strict conclusions. However, each crisis has certain common features and it is possible to learn to some extent from past crises. Not only the shock to the economy and the decline in overall economic activity, but also interest rate measures can be considered common features. Therefore, we present these facts in the article and we have newly added to the discussion. As for the implications, we emphasized them in the Conclusions.

Reviewer´s Comment:

Although your methodology appears clear, I suggest you devote a few lines to research design. I recommend some contributions in the literature to strengthen this section:

- Secinaro, S., Brescia, V., Calandra, D. and Saiti, B. (2020), "Impact of climate change mitigation policies on corporate financial performance: Evidence-based on European publicly listed firms", Corporate Social Responsibility and Environmental Management, Wiley Online Library, Vol. 27 No. 6, pp. 2491-2501.

- Secinaro, S., Radwan, M., Calandra, D., & Biancone, P. (2021). Halal certification impact on firms' corporate social responsibility disclosure: Evidence from the food & beverage sector in Italy. Corporate Social Responsibility and Environmental Management, 28( 4), 1376- 1385. https://doi.org/10.1002/csr.2161

Response:

We would like to the reviewer for this advice. We have modified the section 3, i.e. Materials and Methods. We divided it into the subsection Research Design and Variables and the subsection Data Collection, and we have tried to better refine and explain our research process with added text. We were inspired by the recommended articles.

Reviewer´s Comment:

I suggest providing the discussion section separately from the results. Thus, you can better link it with the satisfactory literature review you have provided.

Response:

We agree with the reviewer that separate sections of results and discussion will be beneficial and therefore, we have made this adjustment and, in addition, we expanded the discussion section.

Reviewer´s Comment:

I suggest you provide theoretical and practical implications in the concluding section. Although they emerge, it does not seem explicitly clear what they are.

Response:

We agree that emphasizing the implications is beneficial and we modified the text in the conclusion section, as already mentioned in the comment above.

Reviewer´s Comment:

The conclusion lacks the research limitations. I would suggest you integrate them.

Response:

We would like to thank the reviewer for this comment, we are aware of the research limitations of our study and therefore, we have added them to the Conclusions section.  

Reviewer´s Comment:

I recommend expanding the section on future research fields, which is essential to continue the ongoing debate on the proposed topic.

Response:

We agree with the reviewer and we added our recommendations for further research at the end of the Conclusions section.

Round 2

Reviewer 1 Report

The authors did a good job of reviewing and correcting the article. Now the article can be published.